# Review on Development and Application of 3D-Printing Technology in Textile and Fashion Design

Ya-Qian Xiao and Chi-Wai Kan *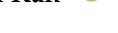

Institute of Textiles and Clothing, The Hong Kong Polytechnic University, Hong Kong, China; yaqian.xiao@polyu.edu.hk
* Correspondence: tccwk@polyu.edu.hk

**Abstract:** Three-dimensional printing (3DP) allows for the creation of highly complex products and offers customization for individual users. It has generated significant interest and shows great promise for textile and fashion design. Here, we provide a timely and comprehensive review of 3DP technology for the textile and fashion industries according to recent advances in research. We describe the four 3DP methods for preparing textiles; then, we summarize three routes to use 3DP technology in textile manufacturing, including printing fibers, printing flexible structures and printing on textiles. In addition, the applications of 3DP technology in fashion design, functional garments and electronic textiles are introduced. Finally, the challenges and prospects of 3DP technology are discussed.

**Keywords:** additive manufacturing; 3D printing; textiles; flexible fabric; fashion design; functional garment

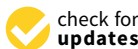



## 1. Introduction

Three-dimensional printing (3DP), also known as additive manufacturing (AM), generates 3D objects rapidly, efficiently and layer-by-layer based on computer-aided design (CAD), computer-assisted manufacturing (CAM) or other digital design tools [1]. Unlike conventional two-dimensional printing, which prints out a two-dimensional pattern presentation or photo plane [2], 3DP can produce complex 3D structures in a shorter cycle and at a lower cost compared to traditional manufacturing. With its high manufacturing efficiency, scalability, low cost and ability to handle complexity, 3DP technology has been applied in many fields, including architecture [3], energy [4], electronics [5], biomedicine [6] and aerospace [7].

In the 3DP process, a product is built by depositing material in successive layers (as if coating layer by layer) until it completed [8]. Three-dimensional printing on textile substrates provides the ability to decorate the fabric's surface without the need for adhesives [9]. Grimmelsmann et al. [10] studied adhesion properties of 3DP materials on textiles and reported that physical "locking" between fabrics and printing materials was the major factor that caused adhesion, rather than chemical bonding. However, adhesion, durability and stability of 3DP onto textiles were found to be difficult to control, limiting the choice of textile substrates and the type of polymer used for 3DP [10]. Moreover, in recent years, 3DP has attracted abundant attention in the textile and fashion industries because of its ability for customization of size and shape. In comparison with the production of traditional cloths, 3DP technology can offer individual customization according to the wearer's body through 3D scanning, and allows for the creation of highly complex structures [11]. However, because of the limitation of raw materials, it is difficult to produce products with the same properties as traditional textiles, in terms of flexibility, scalability and pore structure. Recently, using 3DP technology to develop functional textiles has attracted much attention. Combining 3D printing and textiles offers new opportunities to develop protective equipment and smart textiles while maintaining the comfort and wearability of

fabrics by depositing functional materials onto the textiles. 3DP textiles are likely to find broader applications in the future. Table 1 is a comparison table of 3DP technology and conventional manufacturing technology [12].

**Table 1.** Comparison between 3DP and conventional manufacturing technology [12].

| Advantages | Disadvantages |
|---|---|
| **Flexible Design**<br>— 3DP allows for the printing of more complex designs than traditional manufacturing processes. | **Limited Materials**<br>Selection of 3DP materials is not exhaustive because not all materials can be temperature-appropriate for 3DP. |
| **Rapid Prototyping**<br>— 3DP can manufacture parts within hours, which can speed up the prototyping process and allow for the modifying of designs easily. | **Restricted Build Size**<br>3DP has restrictions on the part size to be printed due to its small printer chamber. For a bigger-sized product, the part may be printed separately and joined together after printing. |
| **Print on Demand**<br>— 3DP does not need a lot of space for stock inventory. | **Post-Processing**<br>Most 3D-printed parts need some form of cleaning-up to remove support material and to achieve a smooth surface. |
| **Minimizing Waste**<br>— The production of parts in 3DP only requires the materials needed for the part itself; therefore, no wastage of raw materials. | **Large Production Volume**<br>Unlike injection molding, 3DP is not cost-effective in a large production volume. |
| **Cost-Effective**<br>— 3DP is a single-step manufacturing process which saves time and costs associated with using different machines for manufacturing. | **Part Structure**<br>3DP generally produces parts layer-by-layer. Although these layers adhere together, they may be delaminated under certain stresses and separated consequently. |
| **Advanced Healthcare**<br>— 3DP is being used in the medical sector to help save lives by printing organs for the human body such as livers, kidneys and hearts. | **Copyright Issues**<br>As 3DP is becoming more popular and accessible, there is a greater possibility for people to create fake and counterfeit products. |

This paper highlights the preparation and application of 3DP for textile and fashion design and also discusses the challenges and perspectives of 3DP in textiles.

## 2. 3D-Printing Technologies in Textile and Fashion Industries

In this paper, 3D printing for textiles is generally divided into four categories (Table 2): (1) extrusion-based 3D-printing systems, including fused deposition modeling (FDM), direct ink writing (DIW) and Electrohydrodynamic direct writing (EHDP); (2) Inkjet 3D printing; (3) Powder bed fusion stereolithography (SLA); and (4) Selective laser sintering (SLS).

### 2.1. 3D Print Method

2.1.1. Material Extrusion

Material extrusion consists of extruding the printing material into filaments using mechanical force and then selectively depositing it through an extrusion nozzle to create a 3D part. According to different printing materials, material extrusion has two common technologies: fused deposition modeling (FDM) and direct ink writing (DIW). The thermoplastic filament is the main element of FDM. The filament is melted into liquid in a heated

liquefier, allowing for raw materials to be smoothly extruded [8]. DIW-printing technology is suitable for materials with special rheological properties, such as hydrogel, elastomer and conductive adhesive. In the printing process, the viscoelastic ink is extruded from the nozzle by pneumatic, piston or screw to form fibers. The electrohydrodynamic direct writing (EHDP) process can be viewed as a near-field model of electrospinning wherein the polymer jet can be deposited precisely. The diameter of the printed fiber can reach the order of nanometers due to the stretching of the electric field. In addition, both polymer melt and polymer solution can be used in EHDP processes.

**Table 2.** A brief description of 3DP technology.

| Techniques | Media | Materials | Advantage | Disadvantage | Ref |
|---|---|---|---|---|---|
| FDM | Solid | Thermoplastics: such as ABS, PA, PLA, PET | Low cost, ease of use, highly available materials | Rough surface | [13,14] |
| EHID | Solid/liquid | PP, PLA, PCL | Low cost, ease of use, soft material, nanofiber | Poor mechanical property, slow build speeds | [15] |
| DIW | Liquid | Polymers, waxes, ceramics | High machine speed, no need of support material | Materials limitation | [16] |
| SLA | | Polyethylene, PP, ABS, polycarbonate, waxes, ceramics. | High resolution, precise and detailed outline, high surface finish | Require support materials, high cost | [9] |
| Inkjet | | PLA, PCL, poly(lactide-co-glycolide) (PLGA), ceramics | Design complex structures | Slow build speeds, materials limitation | [9,13] |
| SLS | Power | Thermoplastics such as Nylon, Polyamide and Polystyrene; Elastomers; Composites | Leftover powders can be reused, freedom of design, durable products | Poor surface finish compared to SLA, high cost | [13] |

### 2.1.2. Inkjet Printing

Inkjet printing is a process in which drops of liquid with diameters measured in micrometers are ejected and thermally deposited to form the part. Droplets in inkjet printing can be generated via two different mechanisms: continuous inkjet printing (CIJ), which continuously produces droplets, and drop-on-demand (DOD) printing, which generates drops when required.

### 2.1.3. Stereolithography Printing (SLA)

Stereolithography, also known as vat photopolymerization (VA), is a laser-based technology that uses a photosensitive liquid resin. The laser (or ultraviolet) beam scans the surface of liquid resin; the resin is then exposed to light and solidified to form a cross-section of the part, and the excess resin is maintained in a liquid state. After a layer is completed, the platform shifts down to a certain height (i.e., the thickness of the printing layer), and solidifies the next layer of resin. The process is repeated layer by layer until the part is prepared.

### 2.1.4. Selective Laser Sintering (SLS)

Selective laser sintering (SLS) uses a laser beam to fuse small particles of powder into a mass that has a desired three-dimensional shape. In the printing process, materials are applied on the bottom of the power bed, and then the laser beam prints according to the pre-designed product shape. Where the laser beam scans, the high temperature of the laser melts the powder and makes the plastic bind together. After a layer is completed, a new layer of materials is applied at the original height, the laser head is scanned again, layer by

layer, until the whole product is completed. SLS is different from SLA and does not need special support structures because the excess powder in each layer acts as a support to the part being built.

### 2.2. 3D-Printing Materials for Textiles

At present, there are various types of 3DP materials, including plastic, resin, rubber, ceramics, gold, platinum, silver, iron and titanium, but not all of these materials are suitable for textile applications. Generally speaking, the 3DP of textiles mainly uses thermoplastic polymers such as acrylonitrile butadiene styrene (ABS), polylactide (PLA), polycaprolactone (PCL), thermoplastic polyurethane (TPU), polyethylene terephthalate glycol-modified (PETG), polystyrene (PS) and polypropylene (PP). Due to the characteristics of the polymers, 3D-printed textiles usually have low air permeability and poor comfort [14,15]. Thus, Wu et al. [16] prepared cotton-containing 3D-printing wires via 2D-braiding technology, which consists of cotton, low-melting polyester (LMPET) and TPU, followed by 3D printing to produce fabric. This cotton-containing 3D-printed fabric shows good flatness and contains a large proportion of cotton powder. Moreover, 3D-printed cotton-containing fabric exhibits better softness, abrasion resistance and tensile properties than polylactic acid 3D-printed fabrics.

## 3. 3D Printing of Textiles

As compared with traditional textiles, 3D-printed textiles are stiff and less comfortable, which restricts the application of 3DP for manufacturing textiles. To solve this problem, several approaches have been carried out to print flexible textiles to improve textile properties including softness, flexibility and stretchability using 3DP technology. Overall, these methods can be classified into three major routes [9]: (1) printing fiber; (2) printing flexible structural units using 3DP techniques; and (3) printing on textiles.

### 3.1. Printing Fiber

Textiles have been widely used as a perfect media in various applications such as thermal management, wearable electronics and energy storage, because of their unique physical properties (e.g., softness, stretchability, scalability) [17–19]. Fiber is the most basic component for textile materials and has many excellent features, including high flexibility, excellent mechanical properties and large surface-area-to-volume ratios [20]. Some researchers have prepared functional fibers through the addition of conductive electric materials via 3DP technology [21–23].

For instance, Wang et al. [24] added lithium iron phosphate/(LFP) and lithium titanium oxide/(LTO) into the PVDF/MWCNT solution, respectively, followed by 3D printing LFP fiber cathodes and LTO fiber anodes, then coated fluoride-co-hexafluoropropylene/ (PVDF-co-HFP) and twisted them together to obtain an all-fiber lithium-ion battery (LIB), as shown in Figure 1a–i. The resultant fibers exhibited high electrochemical performance and good mechanical flexibility, integrated into textile fabrics to develop wearable energy storage (Figure 1a). Similarly, Zhao et al. [25] fabricated fiber-shaped asymmetric supercapacitors and temperature sensors with integrated configuration via inkjet printing, which consisted of $SWCNT/V_2O_5$ fiber cathode and SWCNT/VN fiber anode with coated polyvinyl alcohol. In another study, Cao et al. [26] fabricated flexible smart fibers and textiles via 3D printing with composite inks of 2,2,6,6-tetramethylpiperidine-1-oxylradi-cal (TEMPO)-mediated oxidized cellulose nanofibrils (TOCNFs) and $Ti_3C_2$ 2D transition metal carbides/nitrides. It was noticed that the hybrid inks showed good rheological properties, which allowed them to achieve accurate structures and be rapidly printed.

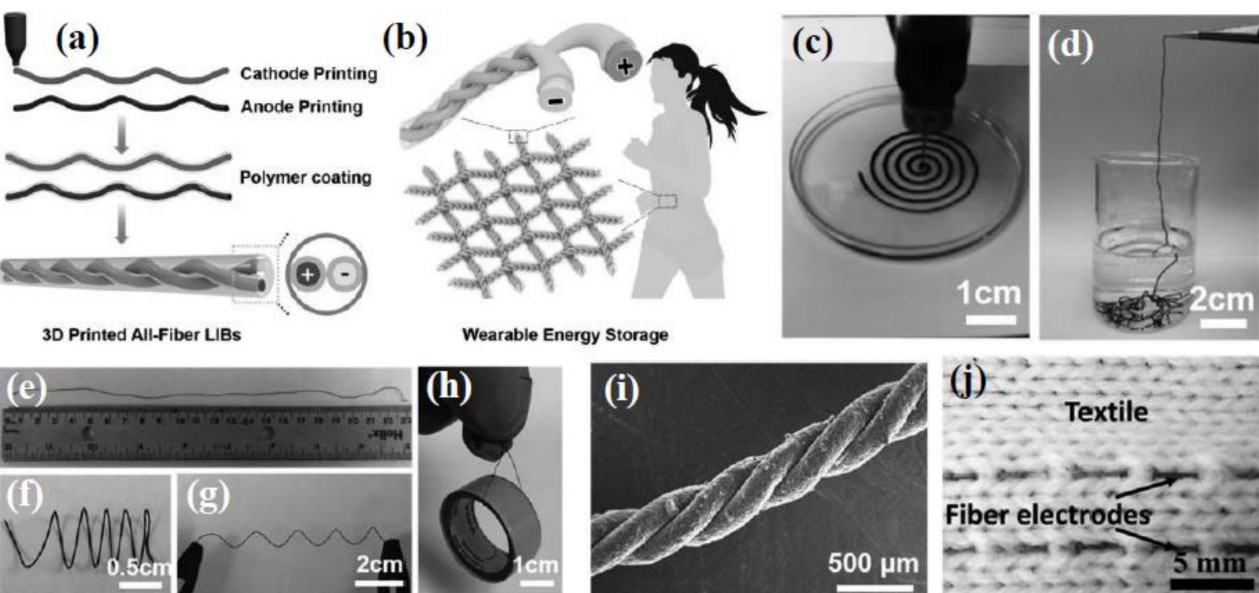

**Figure 1.** (**a**) 3D-printing fabrication process of all-fiber flexible LIB; (**b**) Application of wearable energy storage; (**c**) The optical image of a wet fiber during the printing process; (**d**) The optical image of a wet fiber being removed from the coagulation bath; (**e–h**) The optical image of dry fiber, original fiber, stretched fiber and a fiber bearing a burden, respectively; (**i**) SEM image of composite LFP fibers; and (**j**) LIB fibers are integrated into textile fabrics [26]. Copyright 2017, Wiley-VCH.

### 3.2. Printing Flexible Structures

Many fully 3D-printed garments have already hit the markets in the past few years. Most of these garments are stiff and solid, which are not suitable for people to wear in daily life. These days, various flexible-structure fabrics have been developed, which offer improved, comfortable and practical 3D-printed garments [27].

#### 3.2.1. Traditional Fabric Structures

Inspired by traditional fabric structures, researchers have produced various flexible fabric structures similar to knit or woven structures by 3DP technology [28,29]. For example, Beecroft prepared different types of knit-based structure fabric, including single-layer and tubular structures, using SLS printing of Nylon powder [30]. The obtained fabrics showed great flexibility and extensibility, as with the traditionally knitted structures, as well as mechanical properties of the printed material. In another study, Takahashi et al. [31] developed a new method to produce thin woven fabric using warp (layer-by-layer) and weft (fiber) via FDM 3DP technology. This method could weave a thin fabric through a series of pillars within the design space of the fabricated textile, and the various textiles patterns were prepared by controlling the density of the pillars and weaving patterns. Melnikova et al. [32] developed a lace-structure fabric inspired by Plauen lace, using FDM printing PLA. Recently, Jack Forman from MIT Media Lab introduced a new method with the finely controlled underextrusion of filament to print thin and flexible fabrics named "DefexTiles" [33].

#### 3.2.2. Chainmail Structures

In recent years, designers and researchers have developed 3D-printed chainmail fabrics, which can be bent and folded in the same way as traditional fabrics [34]. Chainmail fabrics are strung together from a series of tiny parts, which can be flexible and customizable according to the parts per unit. For instance, NASA engineers designed a chainmail fabric for protecting spacecraft and spacemen from harsh conditions in deep space. It is flexible and durable and can be changed into various shapes without loss of tensile strength [35]. Designers used a chainmail structure as the base for the fabric and generated a flexible

fabric with different geometric patterns [36]. The University of Hertfordshire and Digital Hack Lab developed a Modeclix structure using SLS 3D printing of PA to create flexible fabrics. This structure contains a system of additively manufactured links, and each link can not only be separated from its neighbors but also be connected, which can create a variety of different patterns and shapes. Thus, garments that are linked by Modeclix can be adjusted to a perfect fit by simply adding or removing links [37].

### 3.2.3. Geometric Structures

Recently, Polymaker and Covestro developed a new method to prepare a 2D fabric by using 3D printing. The new 3D-printing method could create more types of fabrics such as density gradients and moiré patterns than conventional fabric-manufacturing methods [38]. Spahiu et al. [39] designed flexible geometric fabrics via the FDW printing of FilaFlex material. Researchers from MIT produced a flexible and tough mesh fabric inspired by the intertwined structure of collagen using 3D printing TPU [40].

### 3.2.4. Bionic Structures

Bionic design has inspired fashion design for many years. Recently, the use of 3DP technology can help designers to develop bionics in fashion design in faster and more creative ways. The fashion designer would consider biological meanings in fashion design by extracting (i) biomorphic elements, (ii) color elements and (iii) three-dimensional elements and integrating them in fashion design. 3DP allows those ideas to be produced and implemented easily [41].

### *3.3. Printing on Textiles*

Generally speaking, textile products should have flexibility, bending and sufficient tensile strength to adapt to the movement of the wearer's body. However, most printing materials are polymers, and it is difficult for fully 3D-printed textiles to reach these performance requirements. In recent years, some researchers have used 3DP technology for depositing polymers on the surface of textiles to obtain different fabric structures or functional fabrics.

In this process, adhesion between the polymer and substrate plays a key role in determining a material structure's properties. Printing parameters, polymer properties and types of textiles have a great impact on adhesion force [42–44]. Grothe et al. [45] investigated the feasibility of 3D printing resin on different textile substrates, and observed that the textile substrate was too thick and had a smooth surface, which is hard to print onto since the resin does not adhere to it. In another study, Gorlachova et al. [46] discussed the adhesion properties of PLA and Nylon printing on cotton fabric, and found that PLA shows the best adhesion with high printing temperatures and low printing distance, and hydrophobicity polymers exhibit better adhesion than hydrophilic polymers when substrates are made of hydrophilic textile fibers such as cotton. In addition, wear resistance also should be considered for textiles [47]. Eutionnat-Diffo et al. [48] investigated the abrasion resistance of conductive PLA materials deposited onto woven textiles via FDM printing. They reported that the 3D-printed conductive fabric showed a good abrasion resistance in plain-weave fabric, with the highest weft density and the lowest print temperature, and exhibited higher abrasion resistance and lower weight loss after abrasion than the unprinted fabrics.

Rivera et al. conducted 3D-printing experiments on textiles [49]. A new space structure of textile can also be endowed by 3D printing polymers onto the soft fabric. For example, Schmelzeisen et al. [50] used 3DP technology to print structures two-dimensionally on a pre-stressed textile material. After completion of the printing and removal of the pre-stress, the structure changed its extent in the x, y and z directions, which could create complex three-dimensional textures and geometries. Researchers from MIT explored a new method for producing shoes. A two-dimensional pattern was directly deposited onto stretched textiles which was then released after printing, making the shoe jump into pre-designed shapes. This new method could shorten the production cycle of the shoe. In addition, some

researchers have designed various three-dimensional fabrics by a combination of rigid materials with flexible materials, which could be applied in various fields [50–52].

## 4. Applications in the Textile and Fashion Industries

### 4.1. Fashion Design

3DP technology offers freedom of shape, structure and creation for designers; many garments with unique and highly complex structures have been designed, which cannot be produced using conventional textile methods. Generally speaking, 3D-printed cloths lack comfort and flexibility as they are made by polymers, which limits their development in textiles. Therefore, designers and researchers are focusing on the combination of comfort and creativity of garments, and have used a chainmail-like structure to develop soft and flexible apparel [53].

For example, Nervous System studio [54] created a petal dress by SLS printing nylon, which consisted of more than 1600 unique pieces connected by more than 2600 hinges. This dress could be customized to the wearer's body through a 3D scan and each petal could be individually customized including direction, length and shape. In addition, this dress was reconfigurable and could be worn as a top, a skirt or a dress.

3D printing on textiles is a great way to maintain comfort and flexibility of textiles and allow for the creation of highly complex products. Julia Koerner collaborated with STRATASYS to explore digital pattern designs and multi-color 3D printing on fabric inspired by microscopic butterfly wing patterns. First, photographs of the Madagascan Sunset Butterfly's wing setae were digitized into an algorithm and translated into 3D patterns, which corresponded to the form of the garment design. The butterfly pattern was then directly 3D-printed on flexible fabric without any support material [55,56]. Recently, they designed an "ARID collection" that contained a set of 38 3D-printed parts that could be assembled into a full dress or be reconfigured into a variety of ways to form other types of clothing. It was worth noticing that there was no sewing involved in the final assembly of the parts; instead, all seams were joined with 3D-printed joinery [57].

In addition, 3DP technology shows promising applications in jewelry and accessories such as footwear, bags and caps. For example, Acne Studios developed a bag for autumn/winter 2019 using 3D printing technology [58]. So far, using 3D printing technology to manufacture footwear has been a great success and achieved mass production. More and more shoe producers are using 3DP technology to develop products, not only for some parts of footwear, such as the sole, but also for the production of entire shoes. Nike [59,60] developed Nike Flyprint uppers via 3D printing TPU filament. Compared with traditional 2D fabrics, 3D uppers are flexible, lighter and more breathable thanks to an added interconnection beyond the warp and weft.

### 4.2. Functional Garments

Using 3DP technology to create functional garments is another interesting stream of research, because it can create a special textile structure or add functional materials to produce fibers or fabric. Gao et al. [61] prepared a thermally conductive fiber via the 3D printing of poly(vinyl alcohol) (PVA) and boron nitride (BN) to enhance the thermal transport properties of textiles for personal cooling. These fibers exhibited excellent mechanical properties and were used to prepare knitted and woven fabric to develop thermal regulation textiles (Figure 2a). In addition, Pattinson et al. [62] designed a flexible mesh fabric with digitally tailored mechanical properties and geometry via 3D printing TPU. It was found that when waves of the mesh were higher, it was able to stretch more at a low strain before becoming stiffer, which can help to adjust the degree of flexibility of mesh in order to mimic soft tissue (Figure 2b,c).

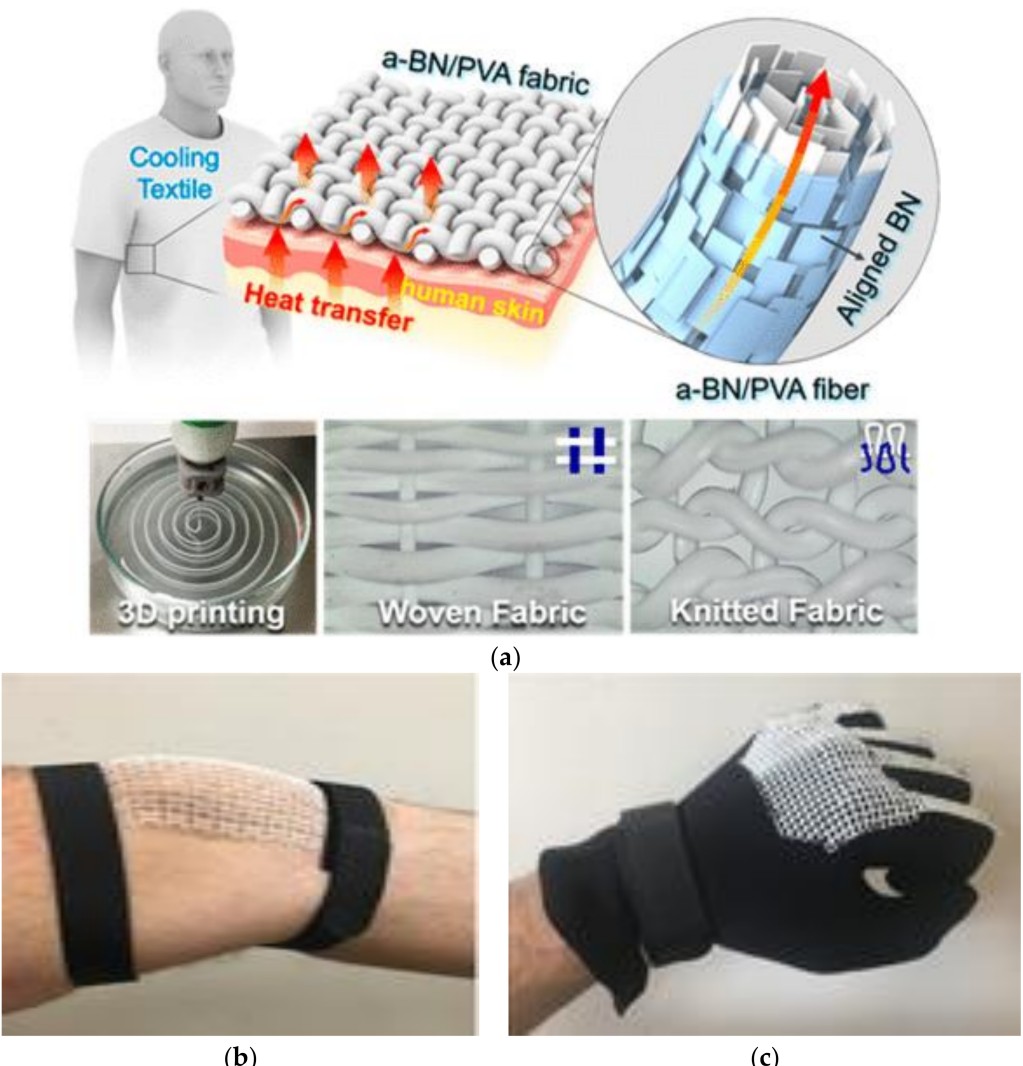

**Figure 2.** (**a**) Schematic illustration of the thermal regulation textile made with 3D printing [61]. Copyright 2017, American Chemical Society. (**b**) A mesh, featuring anisotropic mechanics and showing the ability to conform to a knee. (**c**) The mesh is sewn onto a glove and this mesh-enhanced glove exerts a restoring force on the fingers when the fist is clenched [62]. Copyright 2019, Wiley-VCH.

Recently, Wang et al. [63] designed a structured fabric consisting of two layers of interlocked granular particles using SLS technology, which can gradually switch between soft and hard states. When in its soft state, the fabric can freely bend, fold and drape over curved objects. When the fabric is in its pressured state, the particles interlock and the chainmails jam, and the fabric becomes more than 25 times stiffer than in the relaxed state. In addition, the fabrics showed good shape reconfigurability and were shaped by different geometries. When the fabric was bent into an arch and pressure was applied, the resulting structure was mechanically stiff and could bear mechanical loads more than 30 times its weight. Such tunable and adaptive fabrics have potential applications in areas such as wearable exoskeletons, haptic architectures and reconfigurable medical support.

*4.3. Electronic Textiles*

3DP techniques have been widely used in e-textiles because they can accurately and rapidly print functional complex structures [64–66]. For instance, Zhang et al. [67] fabricated a core-sheath fiber-based smart pattern on textile substrate by ink printing with a coaxial spinneret, which consisted of carbon nanotubes (CNTs) as the conductive core and silk fibroin (SF) as the dielectric sheath, as shown in Figure 3a. The smart textile could harvest

biomechanical energy from human motion and achieve a power density of 18 mW/m$^2$ (Figure 3b). Similarly, Chen et al. [15] prepared stretchable elastic fibers and smart textiles with a coaxial core-sheath structure via 3DP technology, which consisted of graphene as the conductive core and a PTFE insulative sheath. The production of smart textiles can be quickly scaled up when fabricated by 3D printing, avoiding the complexity of the knitting process. The smart textiles achieve the function of a tactile sensor and accurately respond to the contact position and pressure. On the other hand, smart fibers and textiles show excellent stretchability and flexibility and good washability, making them promising for wearable electronics.

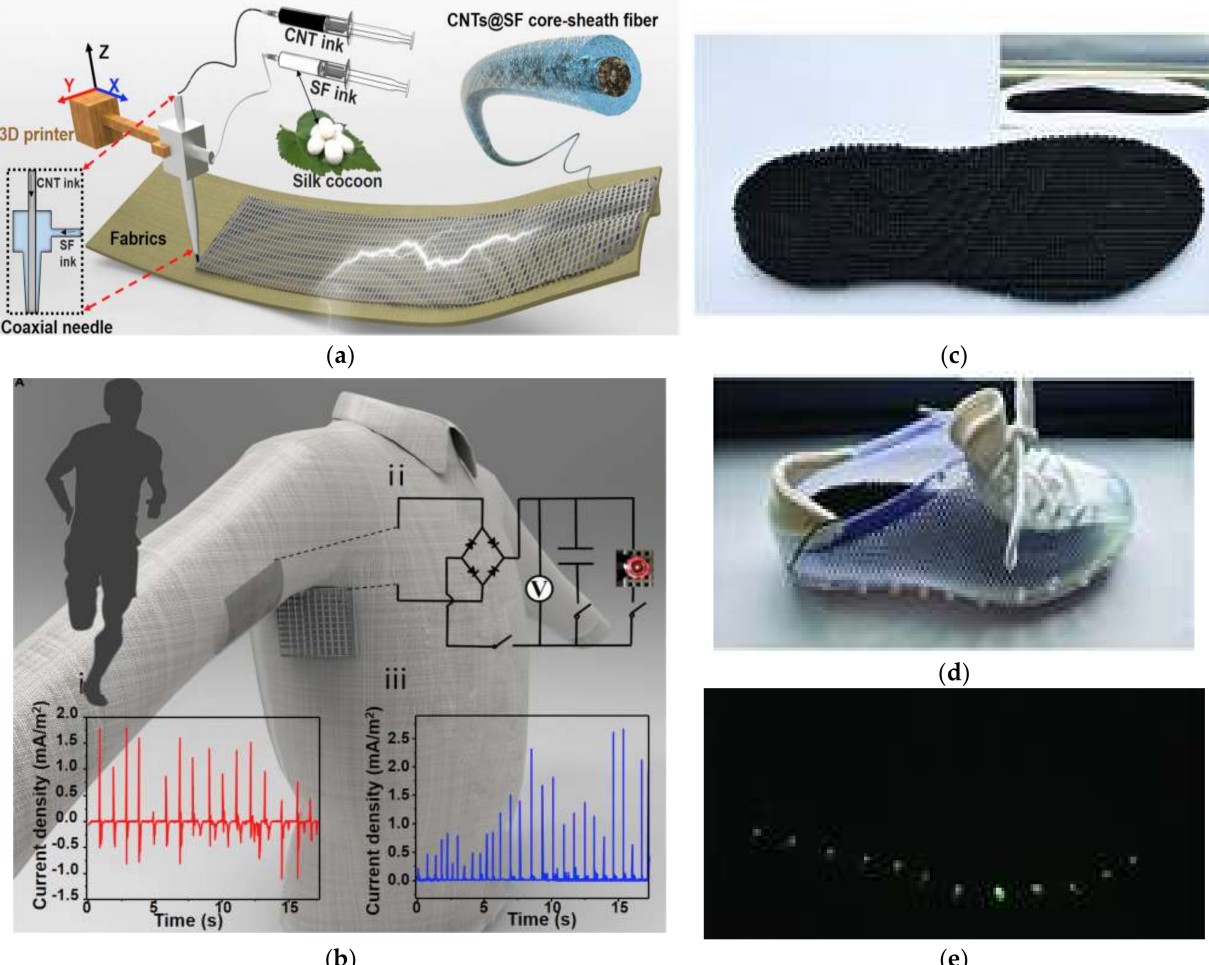

**Figure 3.** (**a**) Schematic illustration of the 3D-printing process using a coaxial spinneret. (**b**) Schematic illustration of smart clothes for energy management and their performances [67]. Copyright 2019, Elsevier Inc. (**c**) 3DP-TENG insole. (**d**) Photograph of a self-powered lighting shoe with a 3DP-TENG insole inside. (**e**) LEDs simultaneously lighting while the wearer stomped [68]. Copyright 2019, Wiley-VCH.

In addition, Chen et al. [68] designed 3D triboelectric nanogenerators (TENGs) with hierarchical porous structure by 3D printing, which consisted of poly(glycerol sebacate) (PGS) as the matrix and conductive CNTs as electrodes. The obtained 3DP-TENG could respond to biomechanical motions and robust energy outputs. It is important to note that TENGs with different shapes and morphologies can be fabricated for various applications because of the high flexibility and controllability of 3D printing. As shown in Figure 3c–e, the 3DP-TENG insole efficiently harvests biomechanical energy to drive electronics.

*4.4. Medical Textiles*

Biofabrication techniques are an interesting usage of textile structures in medical textiles. With the use of 3DP technologies, 3D fibrous scaffolds can be produced specifically to the patient and thus help to facilitate the production of customized tissue structures [69,70]. 3D scaffolds can also provide a template for cell attachment and tissue formation based on the features of conventional woven textile structures [71]. In addition, 3DP can produce heterogeneous structures with programmable Poisson ratios in medical textile applications [72,73].

**5. Challenges and Perspectives**

This review has summarized the recent research on 3DP technology for the textile and fashion industries. 3DP technology offers individual customization and allows designers to make highly complex structures, which is why it has been widely used in textile and fashion design. At present, although great progress has been achieved using 3DP technology for making garments, footwear and some accessories, there are still many challenges that need to be addressed.

First, some designers and scientists have created chainmail structures to produce flexible fabrics, but comfort is still the biggest problem, and there are not many fashion companies that produce garments using 3DP technology. Due to the limit of raw materials, it is hard to achieve pore structures and air permeability such as that of traditional textiles. Therefore, developing new raw materials similar to natural fibers or soft fabric structure is highly desirable. In addition, most 3D-printed wearable garments are currently limited to art pieces or haute couture. These garments involve complex geometric designs and special vivid effects, which always take a long time to produce. It is necessary to design some garments with a low cost that are suitable for daily life. Third, there have been few performance tests of 3DP textiles (e.g., draping, breathability and tensile strength) carried out in previous studies and the unified test standard is also lacking, so the capacities and characteristics of different 3DP textiles are difficult to compare. Therefore, a consensus test method for 3D-printed textile-based structures should be considered.

With the development of 3DP technology and deepening of research, we believe that more materials can be used for 3D printing, and that fabric with comfort and softness can be produced and will be extended in more functional applications.

**Author Contributions:** Conceptualization, Y.-Q.X. and C.-W.K.; methodology, Y.-Q.X. and C.-W.K.; validation, C.-W.K.; formal analysis, Y.-Q.X. and C.-W.K.; investigation, Y.-Q.X. and C.-W.K.; resources, C.-W.K.; data curation, Y.-Q.X.; writing—original draft preparation, Y.-Q.X.; writing—review and editing, Y.-Q.X. and C.-W.K.; visualization, Y.-Q.X.; supervision, C.-W.K.; project administration, C.-W.K.; funding acquisition, C.-W.K. All authors have read and agreed to the published version of the manuscript.

**Funding:** This research was funded by The Hong Kong Polytechnic University for the financial support for this work (Account code: ZJM8 and ZDCC).

**Institutional Review Board Statement:** Not applicable.

**Informed Consent Statement:** Not applicable.

**Data Availability Statement:** Not applicable.

**Acknowledgments:** The authors would like to thank the The Hong Kong Polytechnic University for the financial support for this work (Account code: ZJM8 and ZDCC).

**Conflicts of Interest:** The authors declare no conflict of interest.

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
