# Peer review of "Review on Development and Application of 3D-Printing Technology in Textile and Fashion Design"

_coatings, doi:10.3390/coatings12020267_

Round 1

Reviewer 1 Report

many typo error were detected in manuscript; for example "2.3. D printing technologies".

conventional printing must be discussed and cited

advantages of 3d printing should be identified compared to other coatings technologies.

permissions for figures are the author responsibility.

Author Response

Comment 1: many typo error were detected in manuscript; for example "2.3. D printing technologies".

Reply: The typo was checked.

Comment 2: conventional printing must be discussed and cited

Reply: Addition information was added in “1. Introduction” section and a new Table 1 was added to compare 3DP with conventional technologies.

Comment 3: advantages of 3d printing should be identified compared to other coatings technologies.

Reply: Addition information was added in “1. Introduction” section and a new Table 1 was added to compare 3DP with conventional technologies.

Comment 4: permissions for figures are the author responsibility.

Reply: Figures that cannot find the permission were deleted and permissions of all figures were obtained and attached in the submission system.

Reviewer 2 Report

I think it would be good for the authors to devote a more attention to materials and their structure and comfort properties, and especially special purpose materials. The work would further enrich the part that would also refer to biomimetic materials, because 3D printing is also present in these materials.

Also, it would not be bad to add the part related to medical textiles. My suggestions are aimed at improving the quality of work, because the work will be classified in the group of review papers.
After these corrections, the paper could be published in Journal: Coatings.

Author Response

Comment 1: I think it would be good for the authors to devote a more attention to materials and their structure and comfort properties, and especially special purpose materials. The work would further enrich the part that would also refer to biomimetic materials, because 3D printing is also present in these materials.

Reply: A new section “3.2.4. Bionic Structure” was added in the revised manuscript.

Comment 2: Also, it would not be bad to add the part related to medical textiles. My suggestions are aimed at improving the quality of work, because the work will be classified in the group of review papers.

After these corrections, the paper could be published in Journal: Coatings.

Reply: A new section “4.4. Medical textiles” was added in the revised manuscript.

Round 2

Reviewer 2 Report

I think that this paper could be published in the form that was submitted after the review.